# Tool to Retrieve Less-Filtered Information from the Internet

**Yuta Nemoto \***  **and Vitaly Klyuev**

Software Engineering Lab, University of Aizu, Aizu-Wakamatsu City 9658580, Japan; vkluev@u-aizu.ac.jp
* Correspondence: m5231153@u-aizu.ac.jp

**Abstract:** While users benefit greatly from the latest communication technology, with popular platforms such as social networking services including Facebook or search engines such as Google, scientists warn of the effects of a filter bubble at this time. A solution to escape from filtered information is urgently needed. We implement an approach based on the mechanism of a metasearch engine to present less-filtered information to users. We develop a practical application named MosaicSearch to select search results from diversified categories of sources collected from multiple search engines. To determine the power of MosaicSearch, we conduct an evaluation to assess retrieval quality. According to the results, MosaicSearch is more intelligent compared to other general-purpose search engines: it generates a smaller number of links while providing users with almost the same amount of objective information. Our approach contributes to transparent information retrieval. This application helps users play a main role in choosing the information they consume.

**Keywords:** meta search; news; filter bubble; web development; application



## 1. Introduction

### 1.1. Motivation

Over the last decade, the lifestyles of Internet users have changed dramatically. Methods of communicating with people and gathering information are examples of these remarkable changes. In particular, the popularization of smartphones, involving the spread of social networking services such as Twitter and Facebook, has changed the social system and has had a wide impact on society. While people can access the latest information for free, the risk of spreading incorrect or false information is growing and no longer controllable. It is obvious that people will need to have higher literacy year after year to discern false information from facts when consuming news. Users also need tools to faciliate the autonomic collection of information from multiple sources, and they need to understand concrete facts from diversified perspectives.

As the use of mobile devices has become common, the main sources of news on the Internet have also changed. People prefer to use online media or social media, which can be accessed from their mobile devices, as their news sources. The tendency to use social media has become stronger, particularly for the younger generation. The Pew Research Center [1] summarizes in an investigation of the use of social media news in 2019 as follows: "More than half of U.S. adults get news from social media often or sometimes (55%), up from 47% in 2018" and the "Share of Americans who get news on social media has recently increased". They also report that about half (52%) of all U.S. adults get news from Facebook. According to information from the Reuters Institute [2], the share of social media as a news source has increases over the last nine years. The same article also reports that "Those aged 18–24 (so-called Generation Z) have an even weaker connection with websites and apps and are more than twice as likely to prefer to access news via social media". Young people have grown up in a highly technological environment in which they can have quick access to the latest information anytime, and those online media have begun to occupy a large share of their main news sources. While Internet users have greatly profited from quick information communication, the risk of incorrect or false information

spreading on the Internet is increasing along with the uncertainty and irresponsibility of non-reviewed articles on the Internet. Furthermore, phenomena called "filter bubbles" or "echo chambers" are critical problems in this context.

This situation often attracts attention in the context of politics, especially during an election. The U.S. presidential election conducted in 2016 was one of the most discussed events related to the topic of "Fake News" spreading among users on social networks. Guess et al. [3] estimated that "27.4% of Americans age 18 or older visited an article on a pro-Trump or pro-Clinton fake news website" during the period of the election campaign. These authors found that "social media, especially Facebook, played an integral role in exposing people to fake news". Another article published by BuzzFeed News [4] wrote "the top-performing fake election news stories on Facebook generated more engagement than the top stories from major news outlets" during the campaign. The targets of attention are not only social networking platforms but also search platforms such as Google. Epstein et al. [5] concluded that "the manipulation of search rankings might exert a disproportionately large influence over voters" during political events. Clemons et al. [6] also summarize that "With the ability to manipulate voters even slightly, Google can affect the outcome of a national election". Furthermore, the influence of search engines on politics is discussed on a worldwide level in [7–9]. This means that search platforms have the influence to affect politics. Another fact is that there are people who accuse Google of manipulating search results. CNBC (Consumer News and Business Channel) [10] wrote in a report about the hearing in the U.S. Congress in 2018 that the accusation of manipulation "has been a consistent narrative over the past year, as Republican lawmakers—and even President Donald Trump—have accused Google and other tech platforms of suppressing conservative voices". Google's CEO responded, "Google's search algorithms did not favor any particular ideology, but instead surfaced the most relevant results, which could be affected by the time of a users' search, as well as other factors like their location". The newspaper also reports that one of the Republicans "used sample Google searches to show that Google would turn up positive search results about Republicans and negative search results about Democrats". *The Washington Post* [11] wrote that "Some lawmakers specifically accused Google of weaponizing its popular search engine to put rivals at a disadvantage", and the chairman of the antitrust panel specifically alleged that Google had "stolen content to build [its] own business", citing its practice of culling and displaying information at the top of users' search results in a report about the hearing in 2020.

We do not discuss whether search engines manipulate the search results at a certain level or whether the bias in their results exists. However, because the business of search engines is based on profits from advertisement, it is legal even if they are biased, as reported by CNBC and Reuters [12]. At the very least, the fact that they can influence search outcomes permanently exists. Search engines are also no longer considered an independent instrument in the retrieval of information from the Internet.

Against the background of the spread of misinformation, Vicario et al. [13] concluded that "social homogeneity is the primary driver of content diffusion" and that the tendency of users to take information from a friend with the same profile or people belonging to the same echo-chamber is a part of the mechanisms causing false information to gain acceptance.

Google and Facebook are making efforts to prevent the spread of such misinformation quickly after criticism from society [14]. With regard to the latest situation, Facebook are cooperating with third-party fact-checkers to detect fake news [15] and established a page titled "Voting Information Center" during the 2020 U.S. presidential election campaign [16] to avoid user confusion. However, the process of choosing articles to display on a user's feed and the algorithm that supports users' information acquisition are not open to general users. In particular, topics related to politics and the economy have a huge impact on our society. Tools to help users obtain news autonomously are needed.

*1.2. Potential Vulnerabilities on Social Networks and Search Platforms: Echo Chambers and Filter Bubbles*

Filter bubbles and echo chambers are a part of the serious problems related to social media [17–19]. The phenomenon of a filter bubble is warned to be related to search platforms as well [20,21].

The term "filter bubble" refers to "a situation in which someone only hears or sees news and information that supports what they already believe and like, especially a situation created on the internet as a result of algorithms (=sets of rules) that choose the results of someone's searches" [22]. The word "echo chamber" refers to "a social epistemic structure in which other relevant voices have been actively discredited" [17].

Indriani et al. [23] discussed the positive sides and negative sides of the filter bubble phenomena as follows. They listed the following as the positives: (1) it can prevent people from excessive media exposure; (2) it facilitates our communal instincts; and (3) it can become an assistant for people seeking information because it filters other information that is not needed. On the other hand, the negatives are that (1) people can be stuck in what is called the echo chamber effect, where people seem to think that they know everything; (2) the cause of the filter bubble strengthens the perspective of people who seem to see things in their own perspective; and (3) in some cases, it can benefit advertisements.

Although the effect of the filter bubble is useful for our search behavior, Nguyen [17] pointed out that once people find themselves in the echo chamber, it is psychologically difficult to escape from it. The assumed worst-case scenario for general users regarding these two phenomena is the situation in which people on social media can only gain information filtered by their activities there and they see search results filtered by their thoughts or common behaviors, even when they want to find information from other perspectives.

Because of the situation in which people rely on social networks as their main information source, as introduced above, these phenomena occur on major social networks such as Facebook or Twitter, leading to a risk of immoderate polarization of users or a risk of being drawn into the vortex of "fake news". The risk of this filter bubble problem is not limited to social media but is also common with search platforms such as Google. In this paper, we do not discuss whether these phenomena certainly threaten our society or how much they have affected past events; however, it is an unwavering fact that these social network and search platforms have a closed algorithm to determine the content shown on users' feeds, and practically, their rules or customs can always be changed without being understood by general users.

*1.3. Goal of the Study*

In this paper, we discuss an approach using the concept of a metasearch engine. With the general features of the metasearch engine, it provides search results that do not depend on one summary source, and an individual user's search history or query typing cannot be tracked by the sources. This means the aforementioned problem of the filter bubble is almost disabled with this approach. Nemoto et al. [24] introduced the concept of this method and demonstrated its search result retrieval from multiple search engines based on the logic advocated by Klyuev [25], which are the previous works of the authors of this paper. The goal of this effort is to offer a search system with open logic, no tracking, and less biased search results for general users who want to keep their autonomy to access unfiltered search outcomes. In this paper, we introduce an application named "MosaicSearch" to realize that approach and we conduct an experiment to evaluate its effectiveness.

## 2. Related Works
### 2.1. Overview

Since the US presidential election campaign in 2016, the topic of the "filter bubble", which seemed to affect political events, has attracted the attention of scientists. To develop a clearer picture of this field of study, Amrollahi [26] classified the approaches to this issue

into two categories: studies calling attention to the filter bubble and research works aiming to burst the filter bubble. Amrollahi states that the former contributes to "identification of the bubble, confirming its existence, and quantification of the impact of the bubble" and that the latter is "directly focused on approaches to take users out of the filter bubble". Because our study presents a method to provide users with less-filtered search results, it is classified as a research work that is intended to "burst the filter bubble", according to Amrollahi. In this section, we introduce several other studies that provide the method or tools to support consumers in escaping the filter bubble, works that suggest countermeasures to this problem on search platforms, and some research works from other fields using the mechanism of a "metasearch engine" as practical use cases.

### 2.2. Escaping the Filter Bubble

In this area of research, the approaches allowing users to escape the filter bubble have different characteristics, and it is difficult to systematize them objectively.

In a study similar to our work, Mingkun et al. [27] presented a tool that they developed to "improve people's awareness of their own stances and social opinion selection preferences and mitigate selective exposure". Their system first receives a comment text typed by the user, who inputs their opinion about an article. Then, it analyzes the user's stance against the particular issue and shows some texts picked from a prepared set of comments for the same article. The selected comments include both the same stances of opinions as the user and those with a different stance from the user. Finally, the user selects whether they agree or disagree with each shown comment; then, the system visualizes how the user's opinion is biased to one side of the perspective. They collected a set of articles and comments for these articles from posts submitted by CNN and Fox News on Facebook. After making the collection, they analyzed the opinion of the comments posted for those articles—whether the comment agrees or disagrees with the article—by evaluating the reaction of the author of the comment to the post with an emoticon such as "like" or "love". This enabled users to find opposing opinions, and they concluded that the "system is promising in raising users' awareness of their own stances and preferences, as well as mitigating selective exposure to information".

Although it is not a solution given for textual information, Matan et al. [28] presented automated approaches to identify recorded speech with opposite opinions that counter another specific debate speech. They defined the task in this research as follows: "Given a motion, a supporting speech, and a set of candidates opposing speeches discussing the same motion, identify the opposing speeches recorded in response to the supporting speech". This task was conducted with a set of more than 3600 debate speeches prepared in advance. They evaluated the performance of the task between a case performed by a human and a case performed by some other automated analysis technique. Although they concluded that "expert humans currently outperform automatic methods by a significant margin", it can be a useful approach as a method to provide another perspective to the user if the performance of the automatic method is improved.

Approaches are not limited to solutions on a computer. Foth et al. [29] introduced an approach using civic media on urban screens. They presented some examples of actual projects that are trying to use urban screens "as an opportunity space to provide a platform to help disseminate community and civic information to non-users of conventional digital media who otherwise might not have access or be exposed to such information" and described these approaches as a solution to "break echo chambers and burst filter bubbles". They also wrote that the information posters are not limited to the government or commercial organizations but that information sources can expand to other groups such as nonprofit and community groups, artists, academics, or individual citizens. This approach shows one possibility of a solution beyond user-owned devices.

### 2.3. Countermeasures to the Filter Bubble on Search Platforms

Regarding actual countermeasures to avoid the filter bubble problem on search platforms such as Google, the following ideas were presented. Curkovic [30] suggested that "Logging off from personal accounts, using pseudo-anonymized or advanced search options are options that may reduce this bias" when users are using general search engines. Ćurković et al. [20] recommended "Using speech marks or Verbatim options may reduce automatic reinterpreting of search queries as well as using meta-search engines that operate on different underlying settings (such as DuckDuckGo, Search Encrypt or StartPage)" when users use search engines for scientific purposes. Haddaway et al. [31] suggested an approach by reviewing the storage of scientific data and concluded that their approach substantially increased the search transparency of web-based searches. Indriani et al. [23] introduced several methods to prevent filter bubbles, such as the following:

- reading from other sources to compare whether the information is true;
- clicking on things that are unrelated to what people like and what people usually click on or accessing some information that people might not agree on;
- resetting usernames and starting afresh; and
- reducing the use of social media and not depending on it.

Holoene [21] proposed the following as several possible directions to avoid the filter bubble in the context of health:

- publicizing information about the filter bubble, the hidden algorithms, and the effect that these have on our online lives;
- providing the possibility of unfiltered searching, allowing the public to get unbiased information based on relevance and content quality; and
- switching to another search engine or service such as duckduckgo.com.

Prakash [32] mentioned 10 ways to avoid filter bubbles, such as erasing access history or disabling cookies and personalization on the browser, changing the settings on the social network to keep personal data private, using an incognito browser, and so on.

In this study, we develop a search system that provides search results that are not affected by browsing history, location, or any other user preferences, with an openly accessible logic to select search results, by considering the above suggestions.

### 2.4. Metasearch Engine Mechanism

In this section, we discuss two examples that apply the mechanism of the metasearch engine. Note that these studies show an example of the application of this mechanism, but this is not related to the problem of the filter bubble.

The mechanism of the metasearch engine is at the core of the system proposed in this paper. The term "metasearch engine" refers to a search method that has the following mechanism: "a query is forwarded to one or more third party search engines, and the responses from the third party search engine or engines are parsed in order to extract information regarding the documents matching the query", as written in a patent [33].

Hasan et al. [34] presented their own search system to collect open government data effectively by using the mechanism of the metasearch engine. The novelty of their approach is that the system concentrates on the collection of data published by governments. In particular, they expanded the source search system to retrieve results not only from general-purpose search engines such as Google or Bing but also from government search portals such as data.gov. The output of the search system is the data itself, so the result set is presented in a raw data file (CSV, PDF, XML, etc.). This enables the presentation of more relevant search results. Another notable aspect of this research is that the authors built several modules such as a user preference agent or ordering system to optimize the process to retrieve the search results. For example, the module used to create the search history optimizes and updates the search results for a specific user in combination with another module called the user preference part. Although we do not incorporate this optimization

with the preference of a specific user in this research, their architecture, which combines locally optimized modules to create the overall optimized outcomes, is notable.

Malhotra et al. [35] presented the system of an Intelligent e-commerce website-ranking tool called Intelligent Meta Search System for E-commerce (IMSS-E) by using a combination of the mechanism of a metasearch engine and a big data analytics framework. The purpose of their system was to calculate the rank of a specific e-commerce website to enable an online customer to find a suitable website. The system retrieves various e-commerce websites not from general search engines but from online metasearch engines such as Dogpile, Mamma, Kartoo, and MetaCrawler. An interesting point is that part of their data-collection phase is a metasearch system that retrieves the results from a metasearch engine. The system we develop in this research simply collects the search results from general search engines, but incorporating such secondary search engines to retrieve the results from general engines as a search source is an adaptable method if we can manage the overweighting for search results presented by both secondary search engines and general search engines. They also reduce the number of collected results of websites by calculating factors such as page-loading speed; page freshness; and above all, website relevancy by "calculating frequency of hits for various association rules generated from keywords of search query using Apriori mining algorithm and Map Reduce framework".

In this study, we classify and select results only with the detection of URL domains; however, web page analysis with these factors can be adopted to our approach if we can achieve processing within a limited duration of time.

### 3. Approach

*3.1. Overview*

In this research, we construct a solution mainly for the problem of the filter bubble and aim to offer a search system with open logic, no tracking, and less biased search results, as introduced in Section 1.3. Specifically, we develop an application to provide a search function with these features, which is named "MosaicSearch". The expected scenario stages for this application are as follows: (1) search engines, such as Google, Yahoo, or Yandex, highlight a certain aspect of the information requested by the user via a search query; (2) MosaicSearch collects the search outcomes from these search engines; (3) MosaicSearch reconstructs the search results with an opened method, meaning that the application creates results from the mosaic elements retrieved by the original search engines; and thus, (4) the real nature of the requested information can be understood by the end-user taking these different views into consideration.

To achieve this, we build on the progress reported in our previous work in [24], which discusses the approach based on the logic presented by Klyuev [25]. This approach uses the concept of a metasearch engine and aims to support general users in finding less-biased news related to their needs by retrieving information from multiple sources. The main workflow of this approach is separated into the following four processes: the creation of a collection of search results from multiple sources, the classification of these retrieved documents into predefined categories, the selection of search result items to be presented to the user, and the presentation of these search outcomes to the user. The concept of this workflow is shown in Figure 1. We hypothesize that these processes result in less-biased search outcomes that are balanced in terms of the background of the story. For the process of classification and the selection of result items, we extend the schema presented in these two pieces of research with some adjustments to fit practical development. The main idea of the collection process and the schema that defines classification and selection are presented in the following subsections.

The expected functions that the results in this research can provide to the users are as follows:

- less-biased search result retrieval with a pluralization of search algorithms,
- a limited number of presented search results,
- searches without search/access history,

- search results that do not track the user's personal location or IP address, and
- a mobile-friendly interface useful for general users.

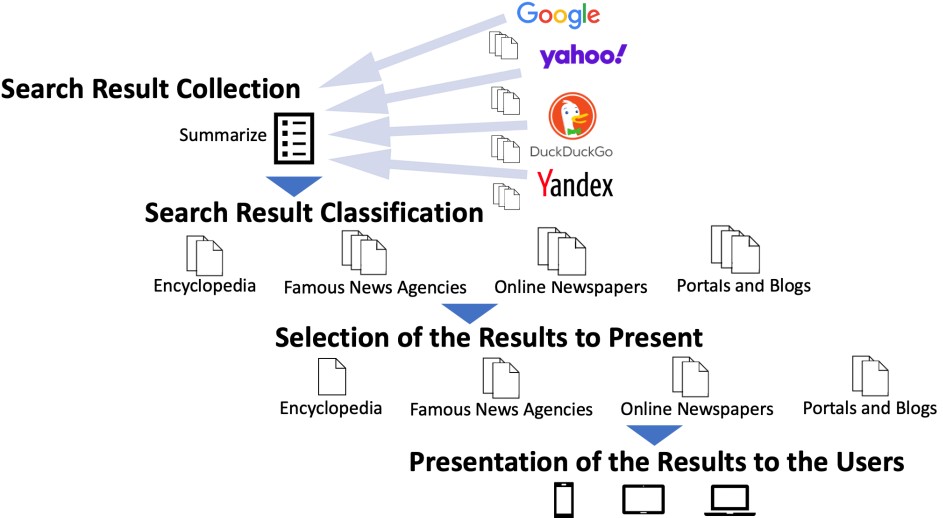

**Figure 1.** Workflow of the retrieval process.

### 3.2. Utilizing Metasearch Engine Mechanism

In this research, we adopted the mechanism of a metasearch engine as a system to reduce the risk of bias in the retrieved search results. In the collection process, we developed a system to retrieve search results from multiple search engines: Google, Yahoo!, DuckDuckGo and Yandex. Because each search engine has its own method of accessing search results, we separated the process to retrieve their search results into several modules that correspond to a specific search engine. After receiving the search results from all search engines in each search module, we collected them into one place, removed duplication by detecting the same article or page using the URL, and allowed them to pass to the next step of classification.

The grounds for using these specific search engines are their global market share, the difficulty to obtain their search results computationally, and the natures of each search engine. For a bigger picture of the candidates of general-purpose search engines to be incorporated into MosaicSearch, see Table 1, which summarizes the search engines listed on the page of Statcounter [36] and NetMarketShare [37]. Regarding Application Programming Interface (API) availability, it is marked as "Available" if existing public APIs offer access to the raw URLs (direct links to the hosts of web pages) of the search result items and the web pages to be indexed on the API service are not crucially different from those of their general services. Regarding scraping difficulty, it is marked as "Easy" if the computational access to raw search results can be performed with scraping based on the constant CSS selector of the search result page. Although the share of Google is dominant in both datasets, Bing and Yahoo! follow as the next group and then Baidu, a Chinese search engine; Yandex, a search engine produced by a Russian company; and DuckDuckGo, a well-known search engine focusing on the user's privacy follow. To select the candidates from which to retrieve search results, Google must be incorporated as the first choice. As Bing and Yahoo! use the same search system to calculate search outcomes [38,39], they offer almost entirely the same or quite similar search results. In other words, we count search results from the same origin if we incorporate both, and thus, only one of Bing and Yahoo! should be included in our search engine. Because it was easier to use Yahoo! in the development process, we selected Yahoo! as a candidate of this source search engine. Baidu has a certain amount of share, especially in the latter dataset; however, we did not include this in our set of search engine candidates because it does not provide any services to retrieve search results, and they make their product structurally difficult to expose to scraping or computational access. We

included Yandex and DuckDuckGo because those brands are ranked at least at the top of market share, and we can retrieve their search result outcomes.

**Table 1.** Candidates for incorporation into MosaicSearch.

| Search Engine | Market Share on Statscounter | Market Share on NetMarketShare | Owner Company | Country of Origin | API Availability | Scraping Difficulty |
|---|---|---|---|---|---|---|
| **Google** | 86.86% | 69.80% | Google LLC | US | Available | Difficult |
| **Bing** | 6.43% | 13.31% | Microsoft Corporation | US | Available | Difficult |
| **Yahoo!** | 2.84% | 2.11% | Yahoo! Inc. | US | Unavailable | Easy |
| **Baidu** | 0.68% | 12.53% | Baidu, Inc. | China | Unavailable | Difficult |
| **Duck-DuckGo** | 0.65% | 0.43% | Duck Duck Go, Inc. | US | Unavailable | Easy |
| **Yandex** | 0.62 (+0.63)% | 1.19% | Yandex LLC | Russia | Available | Difficult |
| **Others** | 1.28% | 0.63% | - | - | - | - |

*3.3. Classification and Selection Processes*

Regarding classification and selection, the search results from MosaicSearch are designed to be from diverse sources. The schema of this process is as follows:

- Only textual data are used as result items;
- Pages are classified into "Encyclopedias" (Wikipedia), "Famous News Agencies" (CNN (Cable News Network), BBC (British Broadcasting Corporation), RT (Russia Today), Al Jazeera, NHK (Japan Broadcasting Corporation), DW (Deutsche Welle), etc.), "Online Newspapers" (*The Guardian*, *The New York Times*, *Financial Times*, etc.) and "Portals and Blogs" by matching the URLs against the list of news agencies and newspapers, which were prepared in advance;
- The maximum number of returned result items to the user is no more than seven, including at least one encyclopedia and two of each other category;
- The application preserves the ranks of result items from the search engine outcomes. If two or more result items have the same rank assigned by search engines, then they are ordered randomly;
- If there are more than two result items from the same source of search engine output, the two with the highest and lowest ranks are selected;
- No more than two representatives from each category are on the final list for the user. One result item has the highest rank, and the other has the lowest. This allows us to increase the chance for result items with different views to be presented to the end user.

Figure 2 briefly presents the schema of the classification and selection process. The three changes from the method presented in the aforementioned studies [24,25] are as follows: (1) the categories of "Portals" and "Blogs" are combined into one; (2) our approach does not provide different criteria based on the device that the user is using; and (3) the scheme to be presented in a PC environment that is assumed in the original research is presented in a mobile device environment as well in this implementation. There were mainly two reasons for these changes: one was the difficulty of implementing an analysis of the page content for purposes such as the classification of "Portals" and "Blogs" or the detection of whether a particular article is the latest news. The use of such a technology to analyze content could be considered in the next step of this research. The other reason was that we decided to implement the search process in a server-side program. As written in [25], the criteria provided to PCs are richer than those for mobile devices in terms of selection quality. Through the use of a server-side program, which can use a large amount of computing resources, the selection schema for the PC environment can be applied without worrying about the latency of the response. The reasons we select a server-side program to serve the search outcomes are that (1) it can use a high amount of computing resources, even in the case of access from a mobile device with poor computing power, and (2) convenience while developing the metasearch process; the use of many libraries for the

development of the retrieval process, such as BeautifulSoup [40]; and efficient processing of the contents in one program.

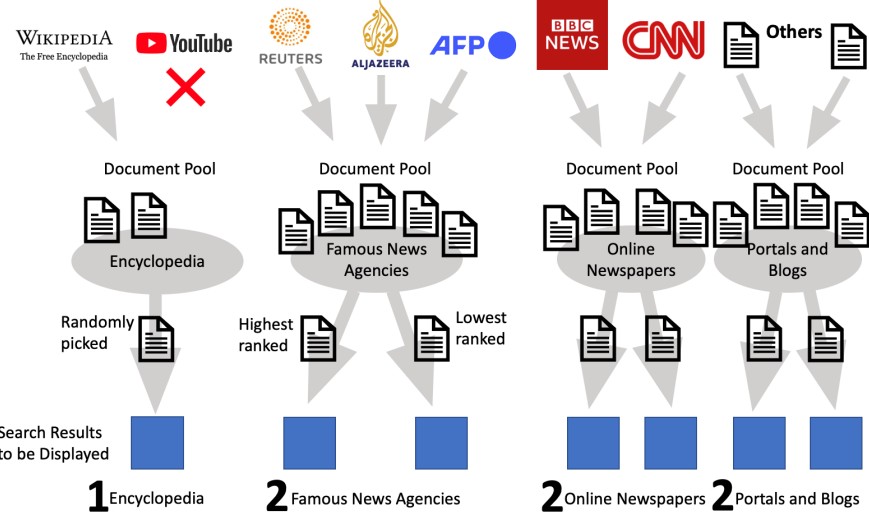

**Figure 2.** Classification and selection processes.

The reason we limit the number of presented search results is because most search system users look only at the first results page. According to Southern [41], the lower the rank of a search result shown on the result page, the lower the possibility that users will click on the result; they wrote that "Unsurprisingly, the tenth position in Google has an abysmal 2.5% click-through rate". This means that users look only at the top-ranked results and search again with other queries if the results are not satisfactory. To develop MosaicSearch, we consider this practice and concentrate the search results to present seven URLs: one from Wikipedia and two from each of the categories "Famous News Agencies", "Online Newspapers", and "Portals and Blogs".

## 4. Implementation

### 4.1. Overview

To implement the approach mentioned in Section 3, we developed a web application that provides search results retrieved from multiple search engines: Google, Yahoo!, Duck-DuckGo, and Yandex. To select the result items to be presented to the users, the application runs four processes: a collection of search results, their classification, a selection of important items, and their presentation to the user. In this section, we discuss the implementation details of each process.

### 4.2. Compiling Search Results

This process is responsible for the collection of search results from multiple search systems. The concrete workflow needed for this process is as follows: (1) throw the query given by the user to each search platform in a predefined format, which depends on the application structure of the specific search system; (2) receive a response from the search engine service; and (3) analyze the received response to summarize the data of search results into a predefined format. Because the method of request and reception of results is different in each search system, we needed to prepare a corresponding program to realize these processes. Thus, these steps were implemented in independent modules that correspond to each search engine and we prepared four modules to collect the search results from the aforementioned search engines. The methods to retrieve the search results from these systems can be roughly grouped in the following two categories: using an API or retrieval by scraping. The former method was applied to Google and Yandex, and the latter was utilized for Yahoo! and DuckDuckGo. Finally, both methods retrieved the search result items' attributes, such as the title of the document, URL, snippets, and rank

in the presented result page; summarized these data; and allowed them to pass onto the classification step.

### 4.2.1. Method with APIs: Retrieval from Google and Yandex

Some of these search platforms officially provide a service to obtain their search results easily. To retrieve search results from Google, we used an API called the Programmable Search Engine [42], which is an official product offered to obtain search results presented by Google. To retrieve the search results for general-purpose searches, we selected the product named "Custom Search JSON API", which allows for implementation on both the client side and server side without ads; however, it has a daily limit for the number of throwable queries. It returns search outcomes in the JSON format, meaning that the process needed for the result-collection function for Google is simply to analyze this returned, well-formatted data. The same type of service is offered by Yandex as well. They offer an API service named Yandex.XML [43], which enables us to access their search outcomes similarly. Some notable differences compared to Programmable Search Engine are (1) the format of the response is an XML format, while Google's response is in JSON format; (2) access to the API from multiple IP addresses is not allowed, so we had to put the access origin in one place.

### 4.2.2. Retrieval Using Scraping: Yahoo! and DuckDuckGo

At the time of writing, Yahoo! and DuckDuckGo do not offer any interfaces to obtain their search results; thus, we used scraping to retrieve their search results. First, we requested their result page and summarized the search result information in a way that corresponds to each search engine. For example, in the case of Yahoo!, a request to the URL "https://search.yahoo.com/search?p=(querytosend)" (accessed on 27 December 2020) drew up their result page. In the case that users gave multiple keywords as the query, it was necessary to combine the words with the character "+". After receiving the page with information from the search results, we used the function of a library named "BeautifulSoup" [40], which is a Python library, to analyze the web page content by specifying elements by tags and attributes in the document, such as the "div" tag of the HTML document or CSS selectors including class or ID. By using this library, it becomes easier to track the place of the aforementioned necessary information such as the page title or URL.

### 4.2.3. Process to Summarize Retrieved Results

Collection of the search results is not complete only with data collection. To pass the data onto the analysis process, some additional processes are needed: removing duplicate items and removing items with videos as their main content. Generally, several search result items referencing the same source and the same article from multiple search engines are retrieved. To specify the ranking of a specific document on the first result page of search engines, we needed to summarize these data and to remove duplicated result items between multiple search engines. Furthermore, after this process, video contents in the search results should be removed. The best way to achieve this is to analyze the page content by accessing each web page. However, the usable time for document analysis before returning a response to the user is limited. To achieve the specification of items, we decided to identify duplication of articles and pages with video as the main content by using the result item's URL. The remaining difficulty is analysis of the duplication of result items which have almost or totally the same content but have different URLs. For example, an article published by Al Jazeera entitled "The 'yellow vest' movement explained" has two URLs: "https://www.aljazeera.com/news/2018/12/04/the-yellow-vest-movement-explained/" (accessed on 27 December 2020) and "https://www.aljazeera.com/news/2018/12/4/the-yellow-vest-movement-explained/" (accessed on 27 December 2020). The only difference between them is the notation method of the date, but our method cannot distinguish between them using the URL as the identifier.

### 4.3. Classification of Retrieved Pages

This process is responsible for the classification of search results into predefined categories: an encyclopedia, a document from a famous news agency, a document from an online newspaper, and portals or blogs. In order to classify the documents, we prepared sets of URL domains that indicate the category of resources of famous news agencies and online newspapers. Retrieved result items are classified into the aforementioned categories based on this set of domains. For example, if the domain of the retrieved result item is "reuters.com", this indicates that the information is published by Reuters, and this item is categorized as a document in the "Famous News Agencies" category. Similarly, if the domain is "nytimes.com", it is regarded as information from *The New York Times* and classified as a document in the "Online Newspapers" category. Figure 3 shows the overall flowchart for the process of classification of search result items and Listing 1 demonstrates the pseudo-code of a practical process for each result item.

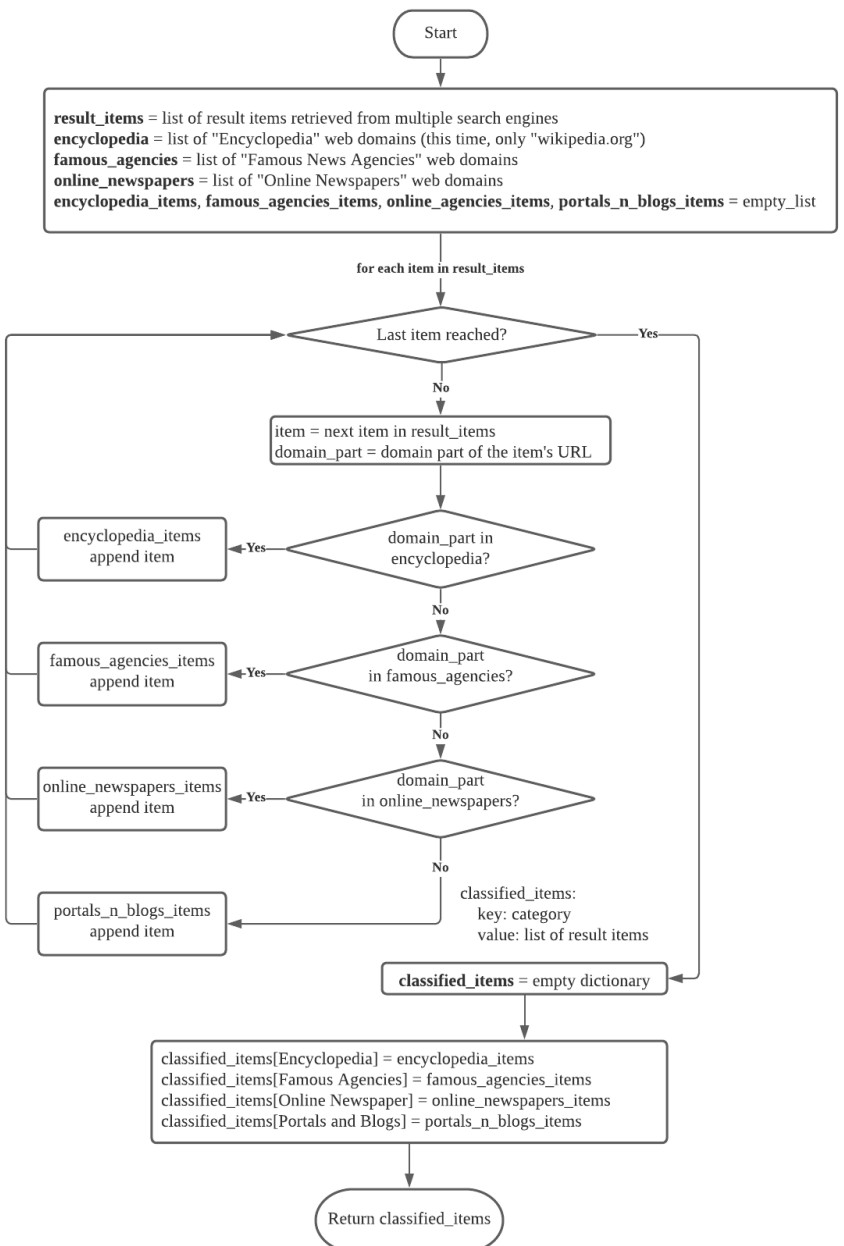

**Figure 3.** Flowchart of the search result classification process.

**Listing 1.** Pseudo-code of the classification process.

```
domain part = domain part of the search result item's~URL

if domain part is ''wikipedia.org'':
list of Encyclopedia items
append the current result item
else if domain part is in
the set of ''Famous News Agencies'' domains:
list of Famous News Agencies items
append the current result item
else if domain part is in
the set of ''Online Newspapers'' domain:
list of Online Newspapers items
append the current result item
else:
list of Portals and Blogs items
append the current result~item
```

4.3.1. Creating the Set of URL Domains

To construct the set of web domains to classify the result items into the predefined categories, we referenced some web resources. To collect information about the domains of the "Famous News Agencies" category, we took the pool of news agencies from a set of names presented by ScienceDirect [44] and a list presented by 4 International Media & Newspapers [45]. To obtain the set of domains of the "Online Newspapers" category, we took the pool of major online news media from a list presented by 4 International Media & Newspapers [46] and another list presented by Ohio University [47]. After that, we omitted unavailable links or renewed the web domains in cases particular domains in the set of URLs were retrieved from these resources to reflect the current availability of these media. These operations were mostly applied for the set of domains "Online Newspapers". For example, one of the candidates of the "Online Newspapers" domain presented by Ohio University's page—the *St. Petersburg Times* with the domain "sptimes.ru"—was no longer available, and therefore, we did not include it in the set of domains to be applied. As another example, the URL of Radio Netherlands' news page with the domain "rnw.nl", which was also presented by the same page, was not accessible. However, this company's website with their news page was available on another domain "rnw.org". In this case, we renewed the domain shown on the source page to the fresh version. One other issue to note is the addition of the domain of a specific media's bilingual page. For example, Aljazeera, an Arabic media company classified as a "Famous News Agencies" domain, has two types of web pages: one includes pages written in the Arabic language (with the domain "aljazeera.net") and the other includes pages written in English (with the domain "aljazeera.com"). Although the page linked from the list presented by 4 International Media & Newspapers is the page in Arabic, we added the domain of the page in English as well to enable users to reach the information published in a more common language.

In summary, the operations we applied to the collection of domains retrieved by the sources mentioned above are as follows.

- Remove the domain from the set if the organization or their website is no longer available;
- Renew the domain if it is no longer available but the same media is available in another domain; and
- Add additional domains if the particular media has a page written in another language.

Finally, our set of domains in the "Famous News Agencies" category included more than 100 news agencies from 69 countries or regions. The set of domains in the "Online Newspapers" category included more than 200 news agencies from 55 countries or regions.

4.3.2. Maintenance of the Set of URLs

Although we prepared a set of URLs that reflected the set of news agencies or news media on the Internet for the experiment, this was not the end of the workflow because the list of media needs to be updated continuously. It is almost impossible to have a comprehensive list of all news agencies or news media all over the world. The perfect coverage of all media in the real world cannot be achieved by our method, especially for the "Online Newspapers" category, because the popularity or the activity of a specific online media source can be changed easily at any time. Thus, we constructed a removal or renewal process for the set of domains, as described in Section 4.3.1. To keep the set of URLs fresh and useful, the workflow used to check the availability or to add new news sources to the list needs to be regularly implemented by considering the changing popularity of media remaining in the set and the existence of other emerging media that have a powerful effect in the real world.

*4.4. Selection of Search Results*

This process is responsible for the selection of search results with the predefined criteria. Through this process, the system selects two to seven results to be presented to the user. The selection is operated by following the criteria defined in Section 3.

The approach for selection of the result items classified as "Encyclopedia" is different from the other categories. First, the operation for the documents in the "Encyclopedia" category is simple: the system selects one "Encyclopedia" result randomly from the set of items classified in this category. Secondly, the process for other categories of result items consists of two stages: removing two or more result items from the same source domain and removing items from categories containing three or more items. Each stage is executed based on the classification result and an analysis of the result items' URL domains resulting from the classification process. Two or more result items are removed from the same domain to decentralize the source of information to avoid retrieval from one biased media source. Similarly, items from categories that have more than three documents are removed to diversify the information source category. We assume that this method can reduce the probability of biased information. In summary, Figure 4 describes the whole process of the selection of search result items, and the program-level design of this process is presented in Listing 2.

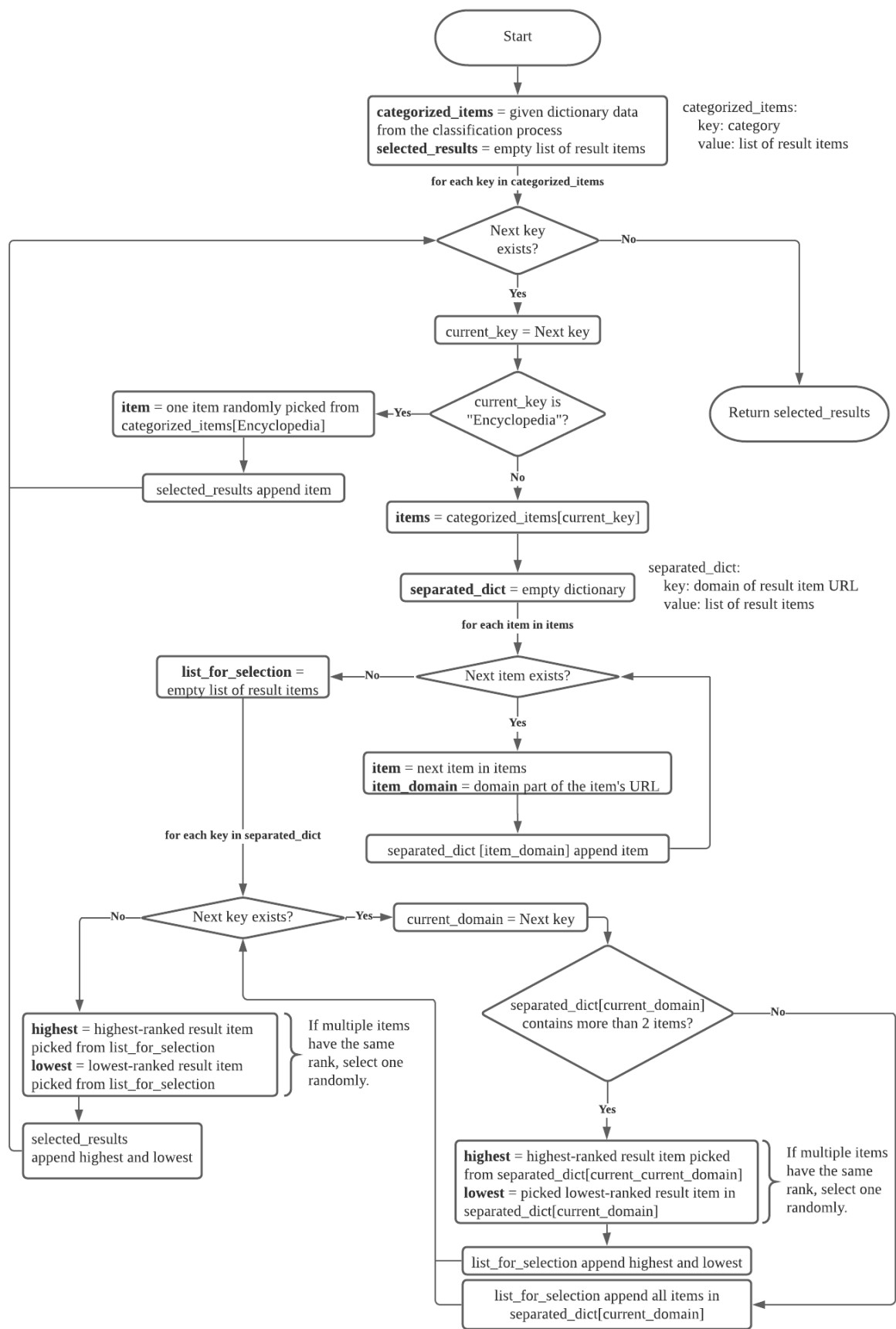

**Figure 4.** Flowchart of search result selection process.

**Listing 2.** Pseudo-code of the selection process.

```
categorized items = {
''Encyclopedia'':
list of result items that are
classified as Encyclopedia,
''FamousNews'':
list of result items that are
classified as a Famous News Agency,
''OnlineNews'':
list of result items that are
classified as Online Newspapers,
''Blogs and Portals'':
list of  result items that are
classified as Blogs and Portals
}

for category in categorized items:
if the category is ''Encyclopedia''
item = pick one item
from the list of this category
the final list of selection append item
else:
items = categorized items content
for the current category
items separated by domains
= separateItemsByDomain(items)
for item list for a domain in separated_by_domains:
list for selection
append the highest-ranked item
from item list for a domain
list for selection
append the lowest-ranked item
from item list for a domain
the final list of selection
append the highest-ranked item
in the list for selection
the final list of selection
append the lowest-ranked item
in the list for selection
return the final list of~selection

separateItemsByDomain(items):
separated_dict = empty dictionary
has a domain as a key, list of items as a~value

for item in items:
items_domain = domain of the item's URL
separated_dict[items_domain] append~item

return~separated_dict
```

### 4.5. Presentation of the Selected Results

This process is responsible for the presentation of search results to the user. The demand for this part of the system is to present the results in a user-friendly format, especially for mobile devices. To achieve this, a responsive web application is applied. The visualization of our product with responsive web design is shown below. Figure 5 shows the content when accessed from a PC, Figure 6 shows the content when accessed from tablet devices, and Figure 7 shows the content displayed on smartphones when the query "hong kong protest" is thrown by the user, respectively.

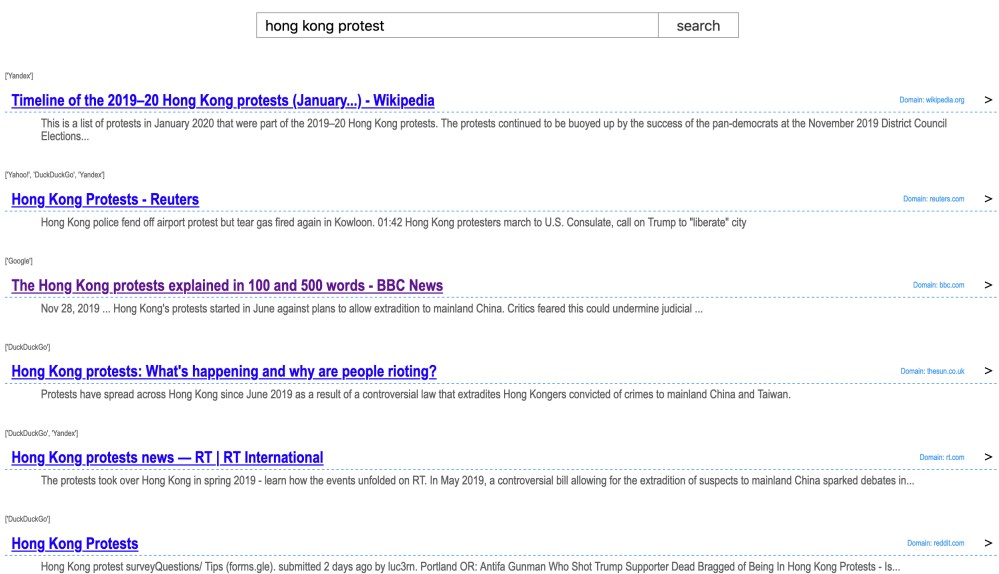

**Figure 5.** Page with search results for PC.

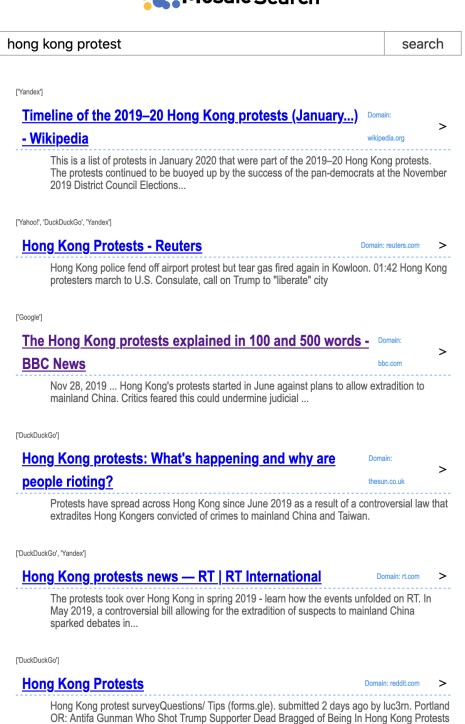

**Figure 6.** Page with search results for tablet devices.

Responsive web design is one of the most basic but important items that should be incorporated into web applications because the number of smartphones accessing the Internet is increasing. According to Statista [48], the number of smartphone users in 2020 was almost 1.5 times greater than the number in 2016. Another source, Pew Research Center [49], reported that "The share of Americans that own smartphones is now 81%, up from just 35% in Pew Research Center's first survey of smartphone ownership conducted in 2011" in their fact sheet published in 2019. Above all, most social network users, who are the largest part of the main target of this product, use mobile devices as to access these services. Business of Apps [50] reported that "96% of Facebook users access the app via a mobile device, with 25% using a computer" in an article published in 2020. Obviously, support for mobile devices is strongly demanded. In this situation, responsive web design affects the usability of the product deeply. Lestari et al. [51] summarized that "responsive web design was able to maintain information quality on home functionality, readability content, and enjoyment of using website but not on information architecture between different mobile browser's sizes" through their experiment. Thus, we implemented this functionality as an important point to improve the user experience. The implementation was carried out by managing the layout according to the screen sizes (width) of user devices. The devices were separated into three groups—PC, tablet devices, and smartphones—and we roughly defined the devices with a viewport width less than or equal to 480 px for smartphones, a width between 481 px to 1279 px for tablet devices, and a width of 1280 px or greater for PC. These are based on the widths of actual devices: the Samsung Galaxy Note 5 or One Plus 3 has a width of 480 px and the Chromebook Pixel has a width of 1280 px, as summarized by IC Web Design [52]. After detecting the viewport width of the access origin device, the system automatically applies the prepared CSS for the specific range of viewport sizes. The CSS is designed by adjusting the content to the representative device's display and is applied as shown in Figures 5–7.

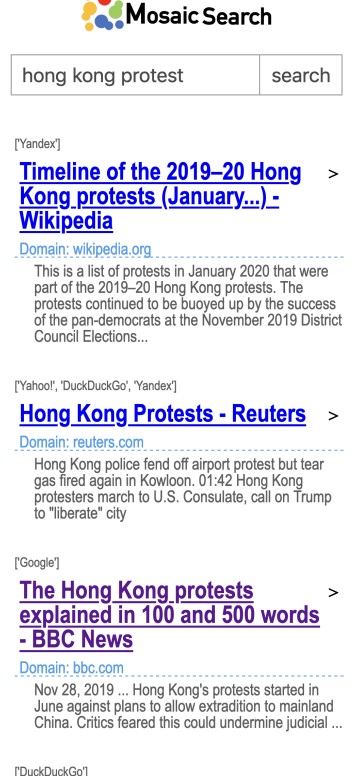

**Figure 7.** Page with search results for smartphones.

*4.6. Implementation and Deployment of the Application in Practice*

The concept and workflow of each process necessary for implementation are presented in the subsections above. In this subsection, we provide brief descriptions of the development framework to implement the application and the method used to make the application public.

For the development of this application, we implemented the system using Python, specifically, the Django framework. There were three main reasons why we selected Django for the development: the amount of development documentation, the required time to build a least-level web application, and the availability of useful libraries. The most important requirement was the last one. A well-known, efficient library for scraping web content, BeautifulSoup, is available on Python, and it is convenient to build a system with a framework on Python to implement search result collection in the application. The source code of this application is available in Zenodo [53].

For deployment of the application, we used a Cloud application-hosting service, PythonAnywhere. In principle, there are mainly two important points to consider before selecting a hosting service: cost (both initial cost and running cost) and required time to build the environment to make the application work. Many cloud computing services have emerged in recent years, and the costs to host web services are reasonable for many services. However, the required efforts to prepare an environment to host an application, such as configuration of the server or installation of a library or framework, are obviously different between services. Regarding this point, almost all of the environment-related preparation can be skipped with PythonAnywhere, and that is why we selected this service to host the application.

## 5. Evaluation and Analysis

*5.1. Method*

### 5.1.1. Overview

The evaluation of the search results' objectiveness or the performance of the system itself was a challenging problem. A novel approach was needed because the traditional approaches—for example, those used in the studies of Kumar et al. [54], Vaughan [55], or Hawking [56]—to measure precision and recall were not appropriate for assessing the objectiveness of the search results. Eventually, to measure the existing bias among the outcomes presented by search systems, a general method was developed to examine the outcomes employing manual judgment by fact-checkers. This experiment also adopted a similar evaluation method.

To analyze the efficiency and usability of MosaicSearch, we conducted an evaluation based on a human-centric assessment. A team of evaluators assessed the performance of this search system by working on the distributed assessment tasks. We present the method used in the evaluation in this section.

The purpose of the evaluation was to assess whether users could obtain objective information related to facts, events, actions, and so on from different viewpoints. In this experiment, we did not collect the evaluator's personal beliefs or preferences for the search result content. The targets of analysis were whether the evaluator could explore an issue objectively with the presented search results and whether the evaluators were satisfied with the presented results obtained through this process.

### 5.1.2. Content of the Evaluation Task

We prepared eight assessment tasks and distributed each of them to evaluators. Each task included a unique search query, the description of the procedure to perform the search, and the questions to assess the quality of retrieved results. Evaluators were asked to answer whether they were satisfied with the presented search results after answering the aforementioned questions. An assessment task for the query was conducted twice: the first time for MosaicSearch, and the second time for the evaluator's favorite search engine. This procedure allowed the performance of MosaicSearch and general-purpose search engines

to be compared. Furthermore, the answers to these questions were collected online via a Google form. The workflow for the evaluation task which the evaluators were required to follow is summarized in Figure 8.

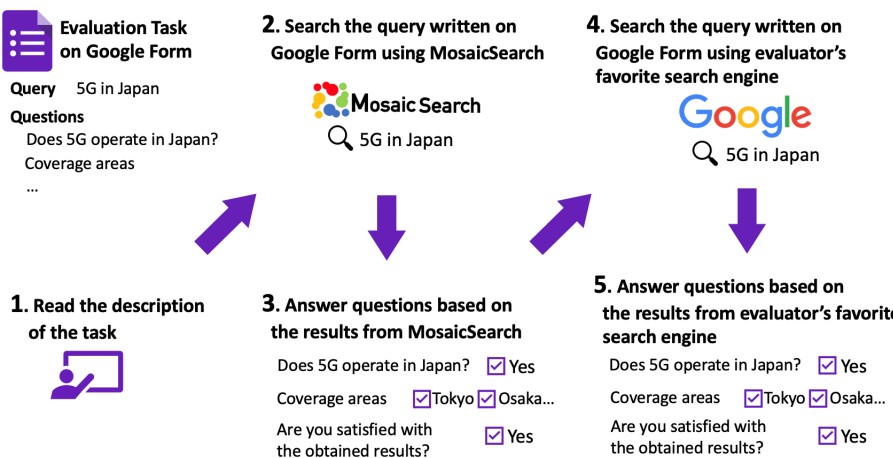

**Figure 8.** Assessor's workflow.

5.1.3. Queries for the Evaluation

We were interested in topics that might be treated differently by countries or companies that own search engines. The following four topics belong to this set: technology, history, politics, and economy. These topics are sensitive. Queries were created in the usual way, i.e., in a manner similar to that used by ordinary users, with two to three keywords or question sentences. We prepared two queries related to each of these topics. Themes such as travel or cooking were not included in the focus of this study because, generally, all search engines will provide equally good results that satisfy the end-users without any effects of filtering on the presented search results. The topics and corresponding queries are shown in Table 2.

Our set of queries related to the most important topics in 2020 from the most common areas of usage of potential users. Regarding queries in technological topics, we focused on 5G and the COVID-19 vaccine, which covered the latest scientific achievements. For queries regarding political topics, we focused on the theme of the U.S. presidential election and the yellow vests movement in France, which were the focus of general activities in politics. Regarding economic queries, we selected a general topic that was expected to interest most people, from two types of perspectives: an overview of the growth and the unemployment rates, which affect their life. In contrast, regarding queries on historical topics, the issues are more stable but those are still discussed, and precise resources from multiple perspectives are required to understand the information objectively.

**Table 2.** Assessment queries.

| Category | Query |
| --- | --- |
| Science | 5G in Japan |
| Science | COVID-19 Vaccine |
| History | Conflict in Cyprus in 1967 |
| History | How big were reparation payments by Germany after the First World War? |
| Politics | Debate 2020 in USA |
| Politics | Yellow vests in France |
| Economy | World economy shrinking in 2020 |
| Economy | Unemployment rate in the world in 2020 |

5.1.4. Set of Questions

The set of queries and the content of the questions are summarized in Tables 3 and 4. The queries in our set may have been understood differently by evaluators. To help them

understand the queries, in the same way, a set of questions was prepared for each query. Most of the questions were answered by selecting the prepared options, such as "Yes/No" or "Specified/Not Specified", but the actual options for the main questions in the distributed evaluation sheet included two extra options: "Information is not presented in the retrieved results" and "This question is outside of my information needs" to avoid a reduction in the quality of answers. These two extra options were omitted in the table.

Most of the questions were used to assess whether a type of objective information was provided. The question set consisted of three groups: questions to assess whether the facts in the text-based information were presented, whether the facts in the number-based information were presented, and whether the information was biased. Examples of the first group were question #1 for the query "5G in Japan" or question #1 for the query "Yellow vests in France". Examples of the second group were question #3 for the query "Conflict in Cyprus in 1967" or question #1 for the query "Debate 2020 in USA". The third group of questions included controversial queries such as "Conflict in Cyprus in 1967" or "Debate 2020 in USA". Question #1 for the former query and question #4 for the latter query were examples of that group. In addition to these main questions, the Google form included a question to ask whether the evaluator was satisfied with the presented search results, and an optional description-type question for any comments related to the system or evaluation procedure.

Regarding the prepared options such as "Specified/Not specified" or "Yes/No", for the questions on Google form, a borderline in the selection of a binary answer was not well-defined because we assumed that whether a particular issue on the question was specified in the content should be judged by the users' information needs.

**Table 3.** Technological and historical queries, questions, and options in the evaluation sheets.

| Queries | Questions | | Options |
|---|---|---|---|
| | **Num.** | **Question Content** | |
| 5G in Japan | 1 | Does 5G operate in Japan? | Yes/No |
| | 2 | Coverage areas (multiple choice) | Tokyo/Osaka/Nagoya/Sapporo/Big Cities |
| | 3 | Date of the coverage of the whole Japan | Specified/Not specified |
| | 4 | Are you as the user satisfied with the obtained results of the search? | Yes/No |
| | 5 | Free way impression (write freely about the quality of the search results) | |
| COVID-19 Vaccine | 1 | Countries developing this vaccine | Specified/Not specified |
| | 2 | Countries registered developed vaccines | Specified/Not specified |
| | 3 | The possible side effects | Specified/Not specified |
| | 4 | Expected date of the vaccine availability in Japan | Specified/Not specified |
| | 5 | Are you as the user satisfied with the obtained results of the search? | Yes/No |
| | 6 | Free way impression (write freely about the quality of the search results) | |
| Conflict in Cyprus in 1967 | 1 | Reasons for the conflict: explanation from both sides are provided | Yes/No |
| | 2 | Casualties during the conflict | Specified/Not specified |
| | 3 | Duration of the conflict | Specified/Not specified |
| | 4 | Results of the conflict | Specified/Not specified |
| | 5 | Are you as the user satisfied with the obtained results of the search? | Yes/No |
| | 6 | Free way impression (write freely \about the quality of the search results) | |
| How big were reparation payments by Germany after the First World War? | 1 | Absolute numbers | Specified/Not specified |
| | 2 | Percentage to entire global GDP | Specified/Not specified |
| | 3 | Duration in years for payment | Specified/Not specified |
| | 4 | Are you as the user satisfied with the obtained results of the search? | Yes/No |
| | 5 | Free way impression (write freely about the quality of the search results) | |

**Table 4.** Political and economical queries, questions, and options in the evaluation sheets.

| Queries | Questions | | Options |
|---|---|---|---|
| | Num. | Question Content | |
| Debate 2020 in USA | 1 | The number of debates between candidates | Specified/Not specified |
| | 2 | Winners in the debates | Specified/Not specified |
| | 3 | Specifics compared to the previous election campaign in 2016 | Specified/Not specified |
| | 4 | Publication expressing the sympathy to only one candidate | Exists/Not exists |
| | 5 | Are you as the user satisfied with the obtained results of the search? | Yes/No |
| | 6 | Free way impression (write freely about the quality of the search results) | |
| Yellow vests in France | 1 | The history of this political movement | Specified/Not specified |
| | 2 | The latest actions are described? | Yes, it's described. /No, it's not described. |
| | 3 | Response of the French government | Specified/Not specified |
| | 4 | Are you as the user satisfied with the obtained results of the search? | Yes/No |
| | 5 | Free way impression (write freely about the quality of the search results) | |
| World economy shrinking in 2020 | 1 | Cause of the shrinking is explained? | Yes/No |
| | 2 | Countries with the worse situations are named? | Yes/No |
| | 3 | Anti-crisis measures are described | Yes/No |
| | 4 | Recovery duration | Specified/Not specified |
| | 5 | Are you as the user satisfied with the obtained results of the search? | Yes/No |
| | 6 | Free way impression (write freely about the quality of the search results) | |
| Unemployment rate in the world in 2020 | 1 | Unemployment rate in different countries, such as United States, China, or Turkey | Specified/Not specified |
| | 2 | Reasons to increase unemployment | Specified/Not specified |
| | 3 | Actions to improve the situation | Specified/Not specified |
| | 4 | Are you as the user satisfied with the obtained results of the search? | Yes/No |
| | 5 | Free way impression (write freely about the quality of the search results) | |

5.1.5. Evaluation Period

The assessment was conducted from the middle of November to the middle of December 2020. Because the search results provided by search engines change day by day, the longer the duration of the experiment, the more different the search results that the evaluators would need to assess. These changes are especially significant for very timely queries such as "Debate 2020 in USA" and the "COVID-19 Vaccine". On the other hand, the challenges are comparatively small for historical queries. We limited the duration of this assessment to a month to maintain a certain level of uniformity of in the evaluators' answers.

*5.2. Result and Analysis*

5.2.1. Evaluators' Backgrounds

The team of assessors consisted of 34 evaluators including 4 professors, 27 university students, and 3 researchers/workers outside of the university. For 11 students, we paid an honorarium. The other 16 students were enrolled in a university course, and an evaluation task was a course exercise for them. The university professors and the remaining evaluators contributed to this assessment voluntarily. Some of them were native English speakers, and some of them were second-language speakers. All of the second-language speakers among the assessors had sufficient ability to read and understand the search results presented in English.

Furthermore, members of the team were not professional searchers such as expert search engine evaluators or fact-checkers. All of them were new to the evaluation at the time of working on this assessment, and we assumed that their experience was closer to that of general users.

We should note that the majority of the evaluators were working or studying in computer-related fields. This means that, when examining the query "5G in Japan", the evaluators may have had better knowledge than ordinary people. For queries belonging to other topics, they did not have professional-level knowledge, although they may have had a personal interest in a specific topic.

Each assessor received only one task because the evaluation process was time-consuming. It took 20 min to 1.5 h to complete the task. This strategy allowed us to minimize possible human errors.

Data on the assessors' favorite search engines were as follows: 31 out of 34 assessors specified "Google", while "Bing", "Baidu", and "Yahoo! Japan" were selected by one evaluator each. Thus, more than 90% of the assessments were compared with Google.

### 5.2.2. Collected Answers

The raw data of the answers collected from assessors after the anonymization of personally identifiable information are available on FigShare [57].

The number of answers submitted to each evaluation task is summarized in Table 5. The number of samples collected in total was 34.

**Table 5.** Statistics on submissions.

| Category | Query | Number of Submissions |
| --- | --- | --- |
| Science | 5G in Japan | 4 |
| Science | COVID-19 Vaccine | 4 |
| History | Conflict in Cyprus in 1967 | 4 |
| History | How big were reparation payments by Germany after the First World War? | 3 |
| Politics | Debate 2020 in USA | 5 |
| Politics | Yellow vests in France | 4 |
| Economy | World economy shrinking in 2020 | 5 |
| Economy | Unemployment rate in the world in 2020 | 5 |

The percentages of the evaluators who answered "Yes" to the question "Are you as the user satisfied with the obtained results of the search?" were 75.21% for MosaicSearch and 82.50% for their favorite search engines, as presented in Figure 9. This indicates that the evaluators' satisfaction level for the search results presented on MosaicSearch was slightly lower than that for other search engines; however, the difference was statistically not significant.

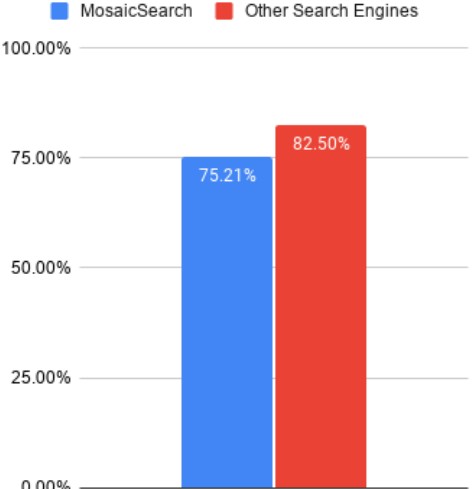

**Figure 9.** Assessors' satisfaction with the search results.

The percentage of evaluators' answers on whether the required information for the question was specified are shown in Figure 10. The data on average across all topics and on eight individual tasks are presented separately for the performance of MosaicSearch and the other search engines. The tasks listed in the figure correspond to the queries in the same order as in Table 2. For example, the query "5G in Japan" was used in task 1, and "Debate 2020 in USA" was used in task 5. Although the evaluators' favorite search engines showed superior performance on average, more than half of the required

objective information for the questions was specified in the search results presented by both MosaicSearch and the assessors' favorite search engine, except for task 5. To obtain a detailed analysis for each query, for the queries in task 2 and task 8—"COVID-19 Vaccine" (62.50% for MosaicSearch and 56.25% for other search engines) and "Unemployment rate in the world in 2020" (78.57% for MosaicSearch and 60.00% for other search engines)—the percentage of specified information on MosaicSearch was higher than that of the evaluators' favorite search engines. For queries for which MosaicSearch had worse performance in retrieving objective information, there were two considerable reasons: an insufficient amount of information because of the smaller number of search results presented or the worse selection of search results because of the randomized choice of result items.

There was no obvious trend between the topics of the queries, and the difference in the performance between MosaicSearch and other search engines in retrieving objective information depended more on the individual queries than on the topic to which the queries corresponded. The data summarized for each topic of the query are presented in Figure 11 and indicate that the search results presented by other search engines have significant superiority in the case of historical topics (the percentages are 64.58% for MosaicSearch and 88.89% for other search engines). This more than 20% difference in percentage for historical queries significantly affects the average. However, the data in Figure 10 show that this topic contained a query that had an excessive difference in the percentage of specified objective information between MosaicSearch and others: the query "Conflict in Cyprus in 1967" on task 3. The difference in another query belonging to the historical topic, which was used on task 4, was around 10%. On the other hand, the difference in the query on task 3 affected the average of historical queries with nearly 40% difference. The same type of analysis can be applied to the political topic as well: the outstanding difference in the case of the query on task 5 overwhelmed the small difference for the query on task 6 and affected the average of the political topic. This means that the significant difference found in Figure 11 was actually not a common tendency between the queries that belong to the same topic. In this comparison, there was no clear relationship found between the topic of the query and the retrieval of objective information.

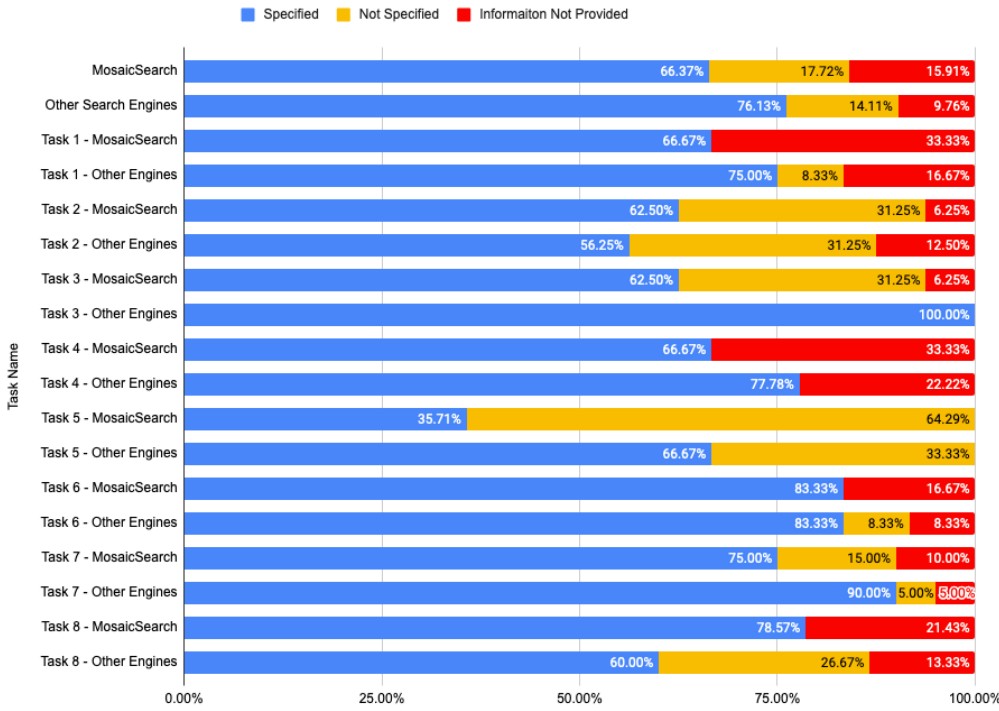

**Figure 10.** Result pages with the specified objective information for individual tasks.

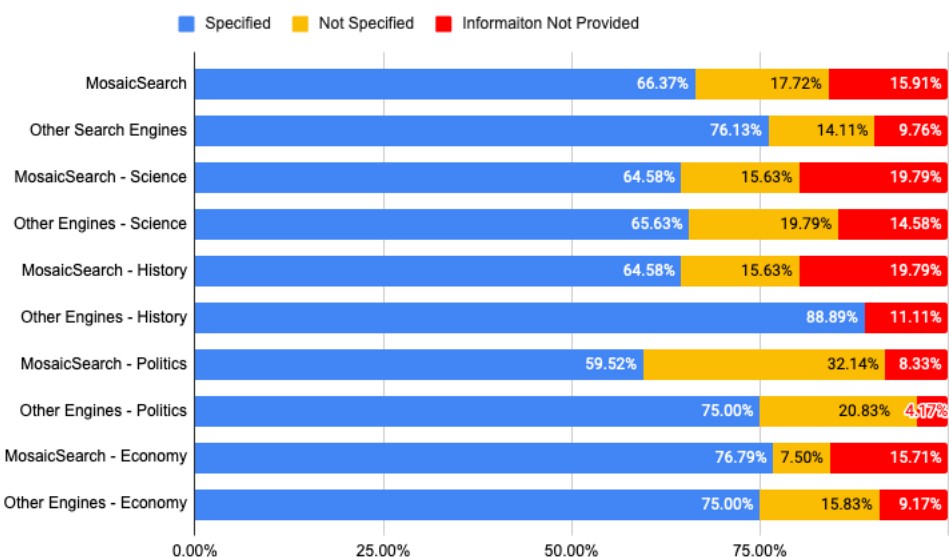

**Figure 11.** Result pages with the specified objective information for individual topics.

The average number of presented search results by MosaicSearch in total was 4.7. The median was 4.71; the maximum number was 7, which was presented for the query "Yellow vests in France"; and the minimum number was 3, which was presented for the query "Conflict in Cyprus in 1967". With our criteria, a larger number of presented results meant that the information sources presented on the search results page of the original search engine were balanced for encyclopedias, famous news agencies, online newspapers, and blogs. The numbers indicate that information from various kinds of sources is presented in the commercial search engine result pages if the number of results presented on MosaicSearch is large, and vice versa.

Regarding the analysis of the results of evaluators' satisfaction levels and the percentage of specified objective information on the result page, MosaicSearch performed sufficiently well when the evaluators searched for and retrieved information related to their query. It seems that the smaller number of search results affected the users' satisfaction levels and the performance of MosaicSearch negatively because of the method of evaluation: evaluators were allowed to search with only one predefined query, although MosaicSearch was designed with the scenario that the users could attempt a search with a new query if they could not find the answer. Generally, the number of search results presented by commercial search engines is within the range of 9 to 15 according to Klyuev [25]. The number on MosaicSearch is smaller compared to that of the other search engines because MosaicSearch focuses on minimizing the search results presented. Although the percentage of retrieved objective information on MosaicSearch is lower than other search engines, in practical use cases, MosaicSearch performs better because nearly 70% of this information is concentrated in the smaller number of results.

Regarding the number of search results, the critical point is that an insufficient number of search results does not provide enough information to compensate for bias if it exists in the search results. Articles/documents including bias towards one side (e.g. republican/democrats and capitalists/socialists) are always present unless the search results are not controlled. In fact, in this experiment, some evaluators answered "No" to the question "Reasons for the conflict: explanation from both sides are provided" on the topic of conflict in Cyprus, and some evaluators answered "Yes" to the question "Publication expressing the sympathy to only one candidate" on the topic of the U.S. presidential debate. Such answers were submitted to both MosaicSearch and their favorite search engines. The fact that bias exists in the presented documents is not a problem; the point is to determine whether an article from a different point of view exists in the search results in the case that a biased article exists. For this point, the smaller number of presented search results on MosaicSearch is a problem. While Google presents 10 search results consistently (in the

default settings), the minimum number on MosaicSearch is 3. The smaller the number of presented search results, the smaller the possibility that MosaicSearch will provide a search result that discusses the issue from another point of view. For controversial queries, a flexible configuration of the number of presentable search results is required.

## 6. Discussion

### 6.1. Advantages and Disadvantages of MosaicSearch

According to the evaluation experiment results, MosaicSearch achieved a sufficient satisfaction level from the assessors and it performed well with the limited number of presented search results.

The advantages of MosaicSearch are the greater amount of objective information per result item provided by the system and the stability of the search performance. It offers a sufficient amount of objective information related to a given query to the users, even with the smaller number of presented search results. The users are not overwhelmed with links. If we suppose that the users take time to read each presented search result equally, the time that users need to take to read all presented search results is almost half that of other search engines. While the time needed to read through the information is reduced by half in MosaicSearch, the difference in performance to retrieve the objective information is small: within 10% on average. Furthermore, this performance rate of MosaicSearch is relatively stable compared to the other commercial search engines because the presented results on MosaicSearch are a mixture of the multiple search engines owned by different companies/countries.

The disadvantages of MosaicSearch include the limited flexibility of the search criteria. While the limited number of presented search results affects the usability of the tool positively, it sometimes produces a lower quality of search results. A lower quality means too much bias in the indexed sources or less relevancy of the presented results. As we mentioned in Section 5.2.2, an insufficient number of links provided on MosaicSearch decreases the variety of perspectives in the search results. It also affects the relevancy of the presented search results. MosaicSearch keeps a certain number of slots for the search results from a specific category even when only one search result is classified in that category. The relativity of the result items is not considered in place of the balance in the information sources. In other words, the limited flexibility of the criteria negatively influences the quality of the search in some cases.

### 6.2. Possible Improvements as a Future Work

To improve the performance of MosaicSearch, the method of configuring the criteria should be discussed in future work. To enable flexible configuration of the search criteria, allowing users to configure this is one possible direction for MosaicSearch. The category types and web domains used to classify the search results may be parameters that can be changed by users. With this function, users can attempt to find suitable criteria to select search results according to their own information needs.

Automated optimization of the source categories or information sources operated by MosaicSearch is not a preferable approach from the point of view of neutrality because the computerized, complex manipulation of criteria generates another filter that the users cannot control. To keep the classification and selection transparent for users, this filtering should be manageable by users.

The method used to assess the performance of the search system also requires improvement. In particular, the problem with the method used in this study is that we were not able to collect a sufficient number of samples for assessment because it was time-consuming for assessors to read all the presented search results and to answer the questions carefully. A considerable improvement in future work would be to evaluate the objectiveness and the quality (relativity) of the search results separately. The former is difficult to evaluate numerically; thus, we conducted a human-centric assessment in this study. However, for the latter issue, there are some methods that could be employed to

perform the measurement without asking many evaluators to make manual judgments on presented pages, such as the calculation of precision and recall, as discussed in other works [54–56], or scoring with methods such as Vector Space Model, Okapi Similarity Measurement, Cover Density Ranking, or Three-Level Scoring Method [58]. In addition, semi-automation of the assessment of search result relevancy is also considerable. An interesting method would be to track the users' activity on the search result page and to measure the relevancy of the result items by their click-through rate [59] or behaviors such as page-scrolling or bookmarking [60]. Incorporating these systems into the process of evaluation would reduce the work and responsibility of assessors and might contribute to increasing the number of samples in the assessment.

## 7. Conclusions

In this paper, we presented an approach to provide less-filtered search outcomes to users. We implemented the MosaicSearch application with the mechanism of the metasearch engine and assessed the effectiveness of the approach. The team of assessors consisted of professors, students at the University of Aizu, and volunteers. We found that the performance of MosaicSearch in retrieving the objective information related to the given query was at a sufficiently usable level. Our evaluation result shows that the percentage of specified objective information presented by MosaicSearch is lower than the other search engines on average; however, MosaicSearch provides a competitive level of search results with a smaller number of search results. While the number of presented search results by MosaicSearch was two times smaller than the other search engines, there was no significant difference between these search systems in terms of performance. On the other hand, the flexible configuration of search criteria should be applied case by case to maintain the quality and neutrality of the presented search results on MosaicSearch. Based on this analysis, we discussed possible improvements as future work.

Providing a way to retrieve search results with an openly accessible logic supplies a certain stability in the neutrality of search results even in situations when commercial searching services impose filters on their search outcomes. Our approach achieved this with a sufficiently useful level of usability. MosaicSearch assists users in understanding the real nature of the requested information, providing them with retrieved documents interpreting events, actions, achievements, etc. from different perspectives. Users may achieve this goal with less effort. This base of the system can be applied to examine other mechanisms by users themselves for further research. In any case, the person who judges whether the retrieved information is correct, useful, or trustful should be the users themselves. The base of the system that enables users to seek a stable method of information retrieval is founded on MosaicSearch. We will continue to improve this system to make further contributions to autonomous information retrieval by users.

**Author Contributions:** Conceptualization, V.K.; methodology, Y.N. and V.K.; software, Y.N.; validation, Y.N. and V.K.; formal analysis, Y.N.; investigation, Y.N. and V.K.; resources, V.K.; data curation, Y.N.; writing—original draft preparation, Y.N.; writing—review and editing, V.K.; visualization, Y.N.; supervision, V.K.; project administration, V.K.; funding acquisition, V.K. All authors have read and agreed to the published version of the manuscript.

**Funding:** This research received no external funding.

**Informed Consent Statement:** Informed consent was obtained from all subjects involved in the study.

**Data Availability Statement:** The evaluation data presented in this study are openly available on FigShare at https://doi.org/10.6084/m9.figshare.13626335 (accessed on 27 December 2020), reference number 13626335. Furthermore, the source code of the main product in this paper is available on Zenodo at https://doi.org/10.5281/zenodo.4460978 (accessed on 27 December 2020), reference number 4460978.

**Conflicts of Interest:** The authors declare no conflict of interest.

**Abbreviations**

The following abbreviations are used in this manuscript:

| | |
|---|---|
| API | Application Programming Interface |
| CSS | Cascading Style Sheets |
| URL | Uniform Resource Locator |
| JSON | JavaScript Object Notation |
| XML | Extensible Markup Language |

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
