# Peer review of "Tool to Retrieve Less-Filtered Information from the Internet"

_information, doi:10.3390/info12020065_

Round 1

Reviewer 1 Report

The main question addresed by this research is that it is possible to provide effective technologies to avoid the "bubble effect" in social networks. With the recent discussion on the formation of the bubble, such solutions are welcome, and must be discussed by the research community. The topic is quite recent, and the paper adds to the area by providing a practical slution to the problem. It would be interesting if it could also provide more performance details.

The conclusions are fine, however, I would like to see more details on the performance.

The article addresses a current topic of great interest to readers in general. The article is well written. I just wish there was a numerical analysis in relation to the reduction of the bias if it were possible.

Reviewer 2 Report

This study presents a metasearch-engine based tool that attempts to avoid the information filtering present in common search engines and thus produce results that avoid pitfalls such as the creation of filter bubbles or echo chambers. The tool is described in detail and a human-centric evaluation method is presented that attempts to measure the tool’s perceived objectivity in relation to common search engines.

The aspect of filter bubbles and the issue of objective information retrieval from the World Wide Web is very important in modern society and is expected to become more and more prevalent in the next decade. This study tackles it with a very pragmatic approach which makes it an interesting take on a very important issue.

A few general remarks:

In the introduction’s motivation section, many examples of articles and research are presented but they seem to be overwhelmingly studying the situation in the USA. It would be a good idea to include similar sources from other areas of the world such as Asia and Europe.

In section 3.2 the method of selecting search engines that are going to be used in the metasearch engine is explained. Adding a comprehensive table of both the selected and not-selected search engines that will display all the relevant information can be very helpful for the reader (e.g., creator/owner company, country of origin, market share, for-profit/non-profit, availability of API/scrapping etc).

In section 3.3 adding a graphic presenting the important parts of the selection process (the 1-2-2-2 distribution rule and the highest-lowest rank rule) will help with readability.

In the “Implementation” section consider using flowcharts to better present the various processes as you see fit. Flowcharts are much easier to comprehend than written text or even pseudocode. The combination of all three will add greatly to user readability.

It might be interesting to make a brief mention of both the software and the hardware tools used to implement and deploy the metasearch engine. Also, you might want to consider providing your project’s full source code alongside the article using a digital object identifier (DOI) if it is open to the public.

Furthermore, consider presenting the evaluation process as described in section 5.1.2 with a simple graphic to help with readability.

As the evaluation results are a cornerstone of this study an effort should be made to showcase them in greater detail.

For example, consider providing the equivalent of Figure 5 for each thematic category (science/history/politics/economy) or even for every query and compare, discuss or draw conclusions for each of them.

Consider providing a different graph with just user satisfaction (only the answers to “Are you as the user satisfied with the obtained results of the search?”) and elaborate on your findings there.

In the section 6.2 where you are discussing future work try to discuss some alternative methods of evaluation that might help to further clarify the differences between MosaicSearch and a classic search. Methods that require less time from the evaluators could be used to increase the number of evaluations and get more trustworthy results, but what could such methods entail? Is there room for some sort of semi-automated or automated evaluation of the result items?

Additionally, a good idea would be to provide all of the questionnaire answers as raw data through a DOI. It might be useful to other researchers.

Concerning some specifics:

When citing previous work from the study’s authors it might be useful to mention it to the reader. This happens in lines 119-120 and again in lines 254-255.

The English text becomes incomprehensible in some parts of the paper. Some extra proofreading could help with that. Some of the lines that have issues are:

Line 25, 86, 262 and lines 306-318 (is the word “document” used to describe result items?)

Overall, the English used in this paper is comprehensible but there is room for improvement in terms of both grammar and phrasing. Some issues were pointed out above but a very thorough proof-reading is necessary.
